# Basic β-1,3-Glucanase from *Drosera binata* Exhibits Antifungal Potential in Transgenic Tobacco Plants

**DOI:** 10.3390/plants10081747

**Published:** 2021-08-23

**Authors:** Miroslav Rajninec, Monika Fratrikova, Eva Boszoradova, Martin Jopcik, Miroslav Bauer, Jana Libantova

**Affiliations:** 1Plant Science and Biodiversity Center, Institute of Plant Genetics and Biotechnology, Slovak Academy of Sciences, Akademicka 2, P.O.Box 39A, 950 07 Nitra, Slovakia; miroslav.rajninec@savba.sk (M.R.); fratrikova.monika@gmail.com (M.F.); eva.boszoradova@savba.sk (E.B.); martin.jopcik@savba.sk (M.J.); 2NAFC—Research Institute for Animal Production Nitra, Hlohovecka 2, 951 41 Luzianky, Slovakia; miroslav.bauer@nppc.sk; 3Department of Botany and Genetics, Faculty of Natural Sciences, Constantine the Philosopher University, Nabrezie mladeze 91, 949 74 Nitra, Slovakia

**Keywords:** antifungal activity, beta-1,3-glucanase, chitinase, hydrolytic enzyme, genetically modified plants, sundew

## Abstract

The basic β-1,3-glucanase of the carnivorous plant *Drosera binata* was tested as a purified protein, as well as under the control of a double *CaMV35S* promoter in transgenic tobacco for its capability to inhibit the growth of *Trichoderma viride*, *Rhizoctonia solani*, *Alternaria solani*, and *Fusarium poae* in an in-vitro assay. The purified protein inhibited tested phytopathogens but not the saprophytic fungus *T. viride*. Out of the analysed transgenic plants, lines 13, 16, 19, and 22 exhibited high *DbGluc1* transcript abundance normalised to the *actin* transcript. Because of *DbGluc1* transgene expression, lines 13 and 16 showed a 1.7-fold increase and lines 19 and 22 showed more than a 2-fold increase in total β-1,3-glucanase activity compared to the non-transgenic control. In accordance with the purified β-1,3-glucanase in-vitro antifungal assay, crude protein extracts of lines 19 and 22 significantly inhibited the growth of phytopathogens (14–34%). Further analyses revealed that the complementary action of transgenic β-1,3-glucanase and 20% higher activity of endogenous chitinase(s) in these lines were crucial for maximising the antifungal efficiency of crude protein extracts.

## 1. Introduction

β-1,3-glucanases (E.C.3.2.1.39) catalyse the hydrolytic cleavage of the β-1,3-D-glycosidic linkages in β-1,3-glucans. In plants, they form multigene families. Particularly, the first investigation of the *Nicotiana tabacum* genome revealed eight genes for acidic β-1,3-glucanases and four to six genes for basic β-1,3-glucanases [1,2]. Later it was shown that *Arabidopsis thaliana* and *Zea mays* genome contain even more complex gene families consisting of 50 and 71 putative genes, respectively [3,4]. Based on structural properties and cellular localisation, they are classified into five (I–V) classes in *A. thaliana*. All sequences contain an N-terminal signal peptide and glycosyl hydrolase family 17 domain. Classes I and IV are vacuolar. Classes I and II have one and class III has two carbohydrate binding modules (CBM43) that are responsible for binding to the substrate [3,5].

β-1,3-glucanases are involved in various plant processes, such as overwintering, inflorescence lengthening, male gametophyte development, pollination, seed development and germination, fruit physiology, defence against biotic and abiotic stress, and nutrient uptake from heterotrophic organisms [6,7,8]. Specific extracellular β-1,3-glucanases bind to the surface of ice crystals, limiting their growth in the apoplast, thus providing cryoprotection [9,10]. They also participate in cleavage of callose surrounding the tetrads and pollen tube growth during pollen development [11,12] and promote rupture of the testa after seed ripening [13]. Moreover, β-1,3-glucanases have received considerable attention due to their role in plant pathogen defence [14]. The first reports associating them with pathogenesis-related (PR) proteins in leaf extracts of tobacco plants revealed the induction of five acidic and three basic isozymes during hypersensitive response to pathogen infection [1,2]. They exhibit different enzyme and antifungal activities [14,15] that are crucial for their role in plant defence. Some of them release pathogen cell wall elicitors that stimulate a defence response [3], cleaving β-1,3-glucan, the structural component of the fungal cell wall. Notably, the diversity of genes for β-1,3-glucanases in the plant genome enabled specialisation toward an active antifungal role during defence. Additionally, these defence hydrolytic enzymes evolved analogues, which in carnivorous plants degrade the trapped prey [16]. 

As β-1,3-glucanases may cleave the β-1,3-glucan substrate present in many phytopathogens with various efficiencies, the individual genes from different organisms were tested in transgenic strategies aimed at improving fungal pathogen resistance [17]. The β-1,3-glucanase gene from saprophytic fungi *Trichoderma* spp. increased resistance in transgenic sugar cane against *Colletotrichum falcatum*, in canola against *Sclerotinia sclerotiorum*, and in rice against *Magnaporthe grisea* [18,19,20]. The β-1,3-glucanases originating from plants exhibited different levels of fungal resistance when they were tested alone or in combination with other defence gene products in transgenic plants. 

The orthologues from tomato, soybean, and tobacco enhanced defence in transgenic mustard, banana, and rice to *Alternaria brassicae*, *Fusarium oxysporum,* and *Rhizoctonia solani*, respectively [21,22,23]. Also, other studies have reported the positive role of transgenic β-1,3-glucanases in resistance against phytopathogens [24,25,26,27,28]. However, prospecting of other genes across a spectrum of organisms is still justified, as individual β-1,3-glucanases may exhibit differences in their antimicrobial potential [27]. Thus far, there is no information in the literature about β-1,3-glucanases from carnivorous plants tested for enhancing resistance against fungal pathogens.

*Drosera binata*, known as fork-leaved sundew, is native to Australia and New Zealand, where it grows in nutrient-deficient habitats. These plants have leaves that function as adhesive traps [29] with the capability to catch and digest prey (mainly arthropods) and subsequently adsorb nutrients [30]. A study of the digestive fluid proteome showed that β-1,3-glucanases are harnessed for the digestion of trapped insect prey cell walls containing β-1,3-glucan [30,31]. Moreover, our preliminary results showed that the isolated gene for an extracellular β-1,3-glucanase class V (QHN63859.1) from leaves of *D. binata* also plays a role in digestion.

In the present study, a β-1,3-glucanase from *D. binata* was cloned and overexpressed in bacterial and plant expression systems to characterise the antimicrobial potential of the purified recombinant enzyme and determine the inhibitory effect of crude protein extracts from transgenic tobacco plants with variable expression of DbGluc1 on a set of fungi. We tested *Trichoderma viride, Alternaria solani*, *R. solani* and *Fusarium poae*. Their cell wall contains 20–30% of proteins, 10–20% of chitin, and 30–55% of β-1,3-glucan, depending on the fungal species [32]. *T. viride* was used as a non-pathogenic test fungus often deployed for characterisation of plant antimicrobial proteins [33], *R. solani* as a tobacco pathogen [34], * A. solani* as a *Solanaceae* crops pathogen [35] and *F. poae* as a rare *Solanaceae* crops pathogen [36]. 

## 2. Results

### 2.1. Antifungal Activity of the Purified rDbGluc1 Protein

To evaluate the antifungal potential of *D. binata* β-1,3-glucanase, the open reading frame lacking a putative signal peptide was amplified using PCR (Appendix A), cloned into the pET32a-Trx vector [37], and introduced into the *E. coli* BL21-CodonPlus(DE3)-RIL expression strain. Following *rDbGluc1* expression and purification of 37.2 kDa recombinant protein (including the tags on both ends) on Ni-NTA agarose, 50 µg of purified protein were supplemented into liquid cultures of filamentous fungi to inhibit the biomass increase and spectrophotometrically tested at 595 nm. The absorbance was measured at the start point (0 h) and the final point of cultivation, which was 24 h for *F. poae* and *T. viride*, 34 h for *R. solani*, and 40 h for *A. solani*. Comparison with the control (heat-treated enzyme) showed that the purified β-1,3-glucanase effectively inhibited the growth of *F. poae* (24%), *A. solani* (14%), *R. solani* (17%) but not *T. viride* (Figure 1; Appendix A).

### 2.2. Production of Transgenic Plants and Molecular Characterisation

A set of transgenic tobacco plants was generated to test the effect of the overexpressed *DbGluc1* gene in crude protein extracts to determine its potential for defence against fungal pathogens. In total, twenty-five transformants were obtained after *Agrobacterium tumefaciens* leaf disc transformation and regeneration on kanamycin selective media. The structure of the T-DNA binary vector pDbGluc1Bin+ is illustrated in Figure 2. The regenerated in-vitro plants showed normal growth and development, similar to control non-transgenic plants. Preliminary screening of the *DbGluc1* transgene transcript level using RT-PCR performed on a set of all regenerated transformants revealed differential transgene expression in individual transgenic plants. To evaluate the contribution of *DbGluc1* expression in transgenic tobacco, the lines (6, 9, 10, 13, 16, 19, and 22) with different (low and high) expression of *DbGluc1* were subjected to complete molecular, biochemical, and antifungal analyses. The presence of *DbGluc1* and *nptII* transgenes was detected by PCR analysis (Appendix A), while the expected size 580 and 500 bp, respectively, of PCR products, was confirmed only in transgenic lines (Figure 3a,b). The *DbGluc1* transgene copy number in individual transformants was estimated using qPCR, and lines 6, 13, 16, 19, and 22 had a single copy. The remaining lines 9 and 10 had two and four estimated copies, respectively (Appendix A).

Unlike the control, all analysed transgenic tobacco lines expressed the *DbGluc1* gene at the mRNA and protein levels. As shown in Figure 3c, the predicted RT-PCR product of 129 bp length was amplified specifically on the RT-mRNA template of transgenic plants carrying the *DbGluc1* transgene. As the *DbGluc1* gene was fused to a short His-tag sequence (encoding six histidines), detection of the DbGluc1His protein in crude protein extracts was performed following separation on SDS-PAGE using the InVision His-tag In-gel Stain. The mature DbGluc1 protein without the first 18 amino acids and with six histidine-tags at the C-terminus consisted of 313 amino-acid residues corresponding to the calculated molecular mass of 34.7 kDa. Figure 3d shows that the specific band of appropriate molecular mass was not detected on SDS-PAGE; instead, a 52 kDa band was revealed in protein extracts of transgenic tobacco lines. A band with a stronger intensity was present in the protein extracts from lines 13, 16, 19, and 22. To confirm that the 52 kDa band was indeed the DbGluc1 protein, the crude protein extracts from transgenic plants and non-transgenic controls were analysed for β-1,3-glucanase activity following separation on SDS-PAGE with laminarin as a substrate. After SDS removal using the Triton treatment, the renatured separated proteins were assayed for β-1,3-glucanase activity. A 52 kDa protein with variable enzyme activity was detected in transgenic lines but not in the non-transgenic control (Figure 3e).

### 2.3. Quantification of the DbGluc1 Expression Level

The seven RT-PCR positive transgenic plants and the non-transgenic control were analysed for relative *DbGluc1* expression through RT-qPCR (Figure 4). The *DbGluc1* mRNA transcript expression level varied substantially among individual transgenic lines. Lines 6, 9 and 10 exhibited a very low mRNA transgene expression normalised to the endogenous *actin* gene level. In contrast, transgenic lines 13, 16, 19, and 22 exhibited 2.7-, 3.6-, 5.9-, and 5.7-times higher *DbGluc1* transcript abundance, respectively, compared to their *actin* transcript (Appendix A).

### 2.4. Quantification of β-1,3-Glucanase Activity

The β-1,3-glucanase activity assay was performed spectrophotometrically at 540 nm with laminarin as a substrate to reveal the contribution of the DbGluc1 enzyme to the total β-1,3-glucanase activity in 100 μg of crude protein extracts isolated from transgenic lines. Non-transgenic tobacco was used as a control.

Transgenic line 6 with extremely low *DbGluc1* expression exhibited β-1,3-glucanase activity that did not significantly differ (*p* < 0.05) from the control, followed by transgenic lines 9 and 10 with very slight (under 20%), and lines 13 and 16 with moderately (~70%) increased enzyme activity. In contrast, the total β-1,3-glucanase activity of lines 19 and 22 was more than 2-fold higher than that in the control (Figure 5; Appendix A).

### 2.5. In-Vitro Antifungal Activity Assay

The hyphal growth of tested filamentous fungi was evaluated spectrophotometrically at 595 nm in the presence of crude proteins (extracted in acetate buffer) originating from transgenic tobacco plants and the non-transgenic control. As shown in Figure 6, none of the protein extracts from transgenic lines suppressed the growth of the saprophytic fungus *T. viride*. Proteins from transgenic lines 6, 9, and 10, with a low *DbGluc1* transgene expression, did not exhibit different antifungal potential compared to the non-transgenic control, in the case of *F. poae*, *A. solani*, and *R. solani*. Surprisingly, four transgenic lines 13, 16, 19, and 22 with relatively high *DbGluc1* transgene expression and increased total β-1,3-glucanase activity (1.7–2.5 times higher than the control) exhibited different antifungal potential. Extracts from lines 13 and 16, whose total β-1,3-glucanase activity was less than 2-fold did not significantly suppress the growth of the tested fungal phytopathogens compared to the non-transgenic control (*p* < 0.05) (Appendix A). In contrast, crude protein extracts from transgenic lines 19 and 22, which had more than 2-fold higher β-1,3-glucanase activity, inhibited the growth of *F. poae* (28 to 35%), *R. solani* (17%), and *A. solani* (14%).

Subsequently, it was determined whether the expression level of DbGluc1 protein or other endogenous hydrolytic enzymes were key factors affecting the antifungal potential of crude protein extracts of tested lines. Since chitin and β-1,3-glucan are synthesised in the apex of growing hyphae simultaneously, hydrolases of both cell wall substrates impact fungal growth inhibition. To determine whether natural tobacco chitinases affected the antifungal potential of transgenic lines, their chitinolytic activities in leaves were analysed and compared to non-transgenic controls. As shown in Figure 5, lines 19 and 22 had increased endogenous chitinase activity (20%; Appendix A). The activity of other stress response enzymes (catalase and ascorbate peroxidase) was not significantly different from the control (data not shown).

These results imply that the antifungal potential of tested transgenic plants depended on *DbGluc1* transgene expression; simultaneously, increased endogenous chitinolytic activity could also significantly contribute to the antifungal potential of lines 19 and 22. A significant growth in chitinolytic activity was detected in two of seven tested transgenic lines that represented lines with more than 2-fold higher total β-1,3-glucanase activity. It can be suggested that a certain threshold of β-1,3-glucanase(s) could function as a trigger signal enhancing endogenous chitinase expression.

## 3. Discussion

Genes for digestive enzymes from carnivorous plants are unexplored potential tools for improving the resistance of plants to pathogens via heterologous expression in transgenic plants. Thus far, only the purified chitinase class I participating in digestive processes from *D. rotundifolia* has been shown to exhibit an apparent antifungal potential against the tested fungi, *F. poae, T. viride,* and *A. solani* [37]. 

β-1,3-glucanases from carnivorous plants with the role in digestion were never tested for this purpose, although they possess hydrolytic activity and are deployed during the decay of insect prey. Such activity is analogous to β-1,3-glucanase isoenzymes specific for defence responses against the attacked pathogen. The in-vitro tests with purified defence-participating β-1,3-glucanases isolated from wheat showed that 35 and 33 kDa isoenzymes suppressed the growth of *R. solani* and *Alternaria* spp., and *F. graminearum* phytopathogens, respectively [38,39]. In contrast, the growth of *T. viride* was not inhibited by β-1,3-glucanases isolated from infected *Pisum sativum* tissue [40]. These results are in accordance with our observation that purified DbGluc1 had the potential to inhibit the growth of *F. poae*, *A. solani,* and *R. solani* but not *T. viride*. Differences in inhibitory effects were described because of several key factors, including the gene structure of plant β-1,3-glucanase, availability of β-1,3-glucan in the fungal cell wall, and production of inhibitors by attacking fungal pathogens [39,40,41,42]. 

Addressing the gene structure, Sela-Buurlage et al. [15] tested defence participating β-1,3-glucanases. Classes I and II are referred to in a new classification [3] as class IV and V enzymes, respectively. Their results showed that only a basic protein class I (class IV) exhibited antifungal potential [15]. Both of these classes of plant 1,3-glucanase contain glyco_hydro 17 domains but differ by the presence (class I/class IV) or absence (class II/class V) of vacuolar localisation signals that do not affect characteristics of the hydrolytic enzyme. 

Strong enzyme and antifungal activities were detected in the case of other basic isoenzymes when purified proteins or transgenic plants overexpressing basic β-1,3-glucanases were tested [21,43,44,45,46].

Here, *DbGluc1*, which contained only the glyco_hydro 17 domain, was tested. *DbGluc1* has basic pI = 8.33, similar to most hydrolases with antifungal potential [15,44]. The basicity of antimicrobial proteins is considered a key factor for antifungal activity because this property enables ionic interactions between the basic protein and negatively charged phospholipid head groups of the fungal cell surface [47,48].

It is known that all tested fungi contain the same major constituents including chitin, and β-1,3-glucans in the fungal cell wall; however, variability in the composition and organization of these polysaccharides [42] may result in the different antifungal effects of individual hydrolytic enzymes. In the case of *Trichoderma* spp., applied chitinases but not β-1,3-glucanases exhibited antifungal activity [15,41,49]. A morphological study of Arlorio et al. [41] revealed that *de novo* synthesized chitin at the apex of the hypha was sensitive to the chitinase resulting in swelling and rupture of the plasma membrane. In contrast, the application of β-1,3-glucanase did not cause swelling and bursting of the hyphal apex. Nevertheless, potential specific inhibitors of β-1,3-glucanase or proteases produced in *Trichoderma* spp. are not the probable reason for β-1,3-glucanase inefficiency, since the combination with chitinase brought synergic antifungal effect [15]. It is more likely, that the subapical glucan-chitin complex became a more accessible substrate for both hydrolases, following the burst of hyphal tips [15,41].

For molecular and antifungal analyses, plants with relatively high *DbGluc1* mRNA expression (3-fold more than the *actin* mRNA level), lower expression, and the non-transgenic control were evaluated. Surprisingly, all transgenic plants yielded the recombinant protein of apparently higher molecular weight (52 kDa) than calculated (34.7 kDa). The same phenomenon was observed in transgenic tobacco with overexpression of sundew chitinase [49]. Even though most of the plant β-1,3-glucanase genes encode proteins with theoretical molecular weights up to 50 kDa [50], crude protein extracts separated on SDS-PAGE gels and analysed for β-1,3-glucanases activity in most cases exhibited a shift toward a higher molecular mass [51]. These differences can be attributed to the post-translational modifications (particularly glycosylation) or (hetero)dimer formation with other cell compounds [52,53].

Lines 13, 16, 19, and 22 with strong expression of *DbGluc1* exhibited higher total β-1,3-glucanase activities than the non-transgenic control. Specifically, lines 13 and 16 showed less than 2-fold higher and lines 19 and 22 showed more than 2-fold higher hydrolytic activity. However, only the crude protein extracts of lines 19 and 22 exhibited the potential to suppress the growth of *R. solani*, *A. solani,* and *F. poae* phytopathogens in accordance with the antifungal activity of the purified DbGluc1 enzyme produced in a bacterial system. The antifungal potential of β-1,3-glucanases may be enhanced by other defence participating hydrolases, including chitinases, which was confirmed in vitro by several studies [15,44,54]. Our results showed that the effect of complementary action of transgenic *DbGluc1* β-1,3-glucanase and endogenous chitinase(s) might be crucial for maximising the antifungal efficiency of transgenic lines 19 and 22. Both lines exhibited increased total chitinolytic activity, reaching 1.2-fold compared to the control. In combination with more than 2-fold higher β-1,3-glucanase activities due to the overexpression of DbGluc1, these tobacco lines achieved antifungal potential capable of inhibiting tested phytopathogens. The enhanced basal expression of some other defence genes, in addition to transgenes, was also observed in genetically engineered plants by other researchers. For example, Dana et al. [55] reported that transgenic plants carrying recombinant chitinase exhibited a higher basal peroxidase activity than the control. Such enhanced enzymatic activity correlated with improved antimicrobial potential of tested plants. The in-vitro antifungal tests performed in our study were focused on the evaluation of the contribution of the *DbGluc1* transgene to antifungal activity. However, the intact transgenic DbGluc1 plants were not analysed for resistance to pathogens. Nevertheless, considering the conclusions of Singh et al. [17], we hypothesised that in the case of in vivo pathogenic infection, tobacco lines 13 and 16 will be more resistant to *F. poae*, *A. solani,* and *R. solani*, similar to lines 19 and 22, due to triggering endogenous PR protein expression.

## 4. Materials and Methods

### 4.1. Plant and Fungal Material

The *N. tabacum* L. (cv. Petit Havana SR1) plants used for genetic transformation were cultured on MS medium (Duchefa, Haarlem, The Netherlands) supplemented with 20 g/L sucrose and 8 g/L agar at 20 ± 2 °C, with a 16 h photoperiod and light intensity of 50 μM/m^2^ s.

The set of filamentous fungi *F. poae*, *T. viride* CCM F486, *A. solani,* and *R. solani* (obtained from Czech Collection of Microorganisms, Brno; http://www.sci.muni.cz/ccm/) (accessed on 23 August 2021) was used for in-vitro antifungal activity experiments. The fungal cultures were maintained on potato dextrose agar (Sigma-Aldrich, Steinheim, Germany) and incubated in the dark at an ambient temperature of 27 °C. Colonies were sub-cultured to Sabouraud agar medium (40 g/L glucose, 10 g/L peptone, 20 g/L agar, pH 5.6) before the hyphal extension assay.

### 4.2. Expression of rDbGluc1 in Escherichia coli and Its Purification

A region coding for the putative mature protein DbGluc1 was amplified by PCR with the *DbGluc1* cDNA as a template, and the primers P1-P2 were designed to introduce restriction sites *Pml*I at the ends of the predicted mature protein-coding region (Appendix A). PCR was performed at 98 °C for 30 s; 98 °C for 7 s, 59 °C for 10 s; 72 °C for 15 s, 35 cycles; and 72 °C for 10 min. The PCR product was digested with *Pml*I, ligated to the pET32a-Trx vector [37], and introduced into the *E. coli* BL21-CodonPlus(DE3)-RIL expression strain (Agilent Technologies, Santa Clara, CA, USA). Expression of the DbGluc1 protein in bacteria was induced by adding 1 mM IPTG to the bacterial culture at an OD_600_ of 0.6, followed by incubation for 3 h at 37 °C. The cells were collected by centrifugation at 4 °C and frozen at −20 °C. The expression of recombinant proteins was checked on 12% (*w*/*v*) SDS-polyacrylamide gel. Purification of His-tagged DbGluc1 from bacterial cultures was performed under native conditions according to the Qiaexpressionist Handbook (Qiagen 2003), with some modifications. The cleared *E. coli* lysate was prepared by two freezing-thawing cycles and resuspended in 5 mL lysis buffer [50 mM NaH_2_PO_4_, 300 mM NaCl, 10 mM imidazole, (pH 8.0)] supplemented with 1 mg/mL lysozyme. Following a 30-min incubation in an ice bath, the samples were sonicated on a Sonicator-150W (Bueno-Biotech, Nanjing, China; 8 pulses for 10 s with 20 s cooling) in lysis buffer enriched with 2% (*w*/*v*) SDS and incubated for 5 min at room temperature. Subsequently, the mixture was incubated for 30 min at 4 °C and centrifuged twice (3700× *g*, 20 min; 4 °C). The supernatant was collected and loaded on a Ni-NTA agarose column (Qiagen, Hilden, Germany) pre-equilibrated with lysis buffer without SDS. Unbound proteins were removed with 20 mL of wash buffer [50 mM NaH_2_PO_4_; 300 mM NaCl; 20 mM imidazole, (pH 8.0)]. Both loading and washing steps were carried out at 8 °C. The recombinant DbGluc1 was eluted at room temperature using 1 mL elution buffer [50 mM NaH_2_PO_4_; 300 mM NaCl; 250 mM imidazole, (pH 8.0)]. Buffer exchange for the purified DbGluc1 protein in the 50 mM *sodium acetate* (pH 5.0) was performed using Econo-Pac 10DG Columns (Bio-Rad, Hercules, CA, USA). Finally, the fresh purified DbGluc1 protein samples were used for the antifungal assays.

### 4.3. Construction of the Plant Expression Vector and Genetic Transformation of Tobacco

The 1132-bp-long sundew (*D. binata*) gene encoding the extracellular β-1,3-glucanase (*DbGluc1*; GeneBank Accession no. MN481115.1) was amplified on the template of genomic DNA with the gene-specific P3-P4 primers (Appendix A). The PCR reaction mixture of 50 µL contained 200 ng DNA template, 20 pmol of each primer, 0.2 mM dNTPs, 1 × PCR buffer, 2.5 mM MgCl_2_, and 2U FirePol DNA polymerase (Solis BioDyne, Tartu, Estonia). The first PCR step was performed at 94 °C for 3 min, followed by 35 cycles of 94 °C for 30 s, 55 °C for 30 s, and 72 °C for 1 min 30 s. The last step was carried out at 72 °C for 10 min. The PCR product was cloned into the pJET 1.2 vector (Thermo Fisher Scientific, Carlsbad, CA, USA) and subjected to Sanger sequencing (Microsynth, Vienna, Austria) (pGlucHis construct).

Two fragments, the d*CaMV35S* promoter isolated from the pCAMBIA1304 vector [56], as *Hind*III–*Nco*I, and the *DbGluc1* gene from the pGlucHis construct, as *Nco*I–*Xba*I, were ligated into the pBinPLUS vector with a cloned *35S* terminator [41] and digested with *Hind*III–*Xba*I enzymes.

Subsequently, the pGlucHisBin+ construct (Figure 2) was introduced into *Agrobacterium tumefaciens* strain LBA 4404 and used to transform leaf discs of tobacco following the protocol described by Mlynarova et al. [57]. Regenerated transgenic shoots were rooted in solid MS medium supplemented with 20 g/L sucrose, 50 mg/L kanamycin, and 500 mg/L cefotaxime (Duchefa, Haarlem, The Netherlands). 

### 4.4. PCR Analysis

To verify the presence of transgenes in the putative DbGluc1 transgenic plants, genomic DNA was isolated from leaf tissue of individual transgenic lines and the non-transgenic control using the protocol of Chen et al. [58] and was subjected to PCR. Primers P5-P6 and P7-P8 were used to confirm the presence of *npt*II and *DbGluc1* expression units, respectively (Appendix A). The 25 µL PCR reaction mixture contained 100–200 ng DNA template, 10 pmol of each primer, 0.2 mM dNTPs, PCR buffer, 2.5 mM MgCl_2_, and 1 U FIREPol Taq DNA polymerase (Solis BioDyne, Tartu, Estonia). PCR was performed at 94 °C for 2 min; 94 °C for 30 s, 56 °C for 30 s, and 72 °C for 1 min for 35 cycles; and 72 °C for 10 min. The PCR products were separated on a 1% (*w*/*v*) agarose gel and visualised with ethidium bromide staining.

### 4.5. Copy Number Estimation by qPCR

The transgene copy number was determined by qPCR by the Pfaffl method [59] as the ratio between normalised amounts of the β-actin reference gene and transgenic glucanase gene (*DbGluc1*), as the target. Plants with a single copy number confirmed by Southern blot were used as calibrators. Quantitative PCR was performed using the primers P9-P10 and P11-P12 (Appendix A), LightCycler Nano (Roche Applied Science, Penzberg, Germany), and 5 × HOT FIREPol EvaGreen mix (Solis Biodyne, Tartu, Estonia) according to the manufacturer’s instructions. The cycling conditions consisted of incubation at 95 °C for 12 min, followed by 40 cycles of 95 °C for 10 s, 60 °C for 30 s, and 72 °C for 30 s. Melting curve analysis was performed at 60–97 °C. The experiments were repeated three times.

### 4.6. RT-qPCR

Total RNA was isolated from the leaves of in-vitro cultivated transgenic plants and the non-transgenic control using the RNeasy Plant Mini kit (Qiagen, Hilden, Germany) according to the manufacturer’s instructions. First-strand cDNA was synthesised using the Maxima First strand cDNA Synthesis kit for RT-qPCR (Thermo Fischer Scientific, Vilnius, Lithuania).

RT-qPCR was performed using the Luminaris HiGreen qPCR Master Mix (Thermo Fischer Scientific, Vilnius, Lithuania) according to the manufacturer’s recommendation with the primers P9-P10 (tobacco actin, XM_016618658.1) and P11-P12 (*D. binata* glucanase, MN481115.1) (Appendix A). The reaction was initiated by an uracil-DNA glycosylase step at 50 °C for 2 min, followed by one cycle at 95 °C for 10 min. The reaction proceeded for 40 cycles at 95 °C for 15 s and 60 °C for 60 s and was completed with a melting curve analysis step to confirm the specificity of amplified products. A LigthCycler Nano (Roche Applied Science, Penzberg, Germany) was used for qPCR. Experiments were performed in biological triplicate (three independent isolations, reverse transcriptions, and qPCR reactions for each plant) and technical duplicate, and the threshold Ct value was set according to the automatic calling method and melting curve analysis of the associated software. The Pfaffl method was employed to process the relative gene expression data [59]. Briefly, values were expressed as a ratio between the target (transgenic *DbGluc1*) and a housekeeping gene (*actin*). The correct primer efficiency (E) was calculated using a standard curve obtained from qPCR reactions using a 5-point 5-fold diluted cDNA template. The target gene expression ratio was then calculated with the Pfaffl equation: ratio = (E_target_)^Cttarget^ ÷ (E_actin_)^Ctactin^.

### 4.7. Fluorescent Detection of the His_6_-Tag Sequence of DbGluc1 Recombinant Protein

Total proteins were extracted from transgenic plants and a non-transgenic control using extraction buffer containing 0.05 M sodium acetate (pH 5.2) and 0.02% (*v*/*v*) β-mercaptoethanol. Aliquots (30 μg) were mixed with a 2 × loading buffer [0.09 M Tris, pH 6.8; 20% (*v*/*v*) glycerol; 20 g/L SDS; 0.2 g/L bromophenol blue; and 0.1 M DTT], heated at 100 °C for 5 min, and separated on 12.5% mini-gels (Mini-Protean Cell, Bio-Rad, Hercules, CA, USA) according to Laemmli [60]. 

Subsequently, the gels were fixed in a 200 mL solution containing 10% (*v*/*v*) acetic acid and 50% (*v*/*v*) methanol for 1 h, washed twice with ultra-pure water for 10 min, and incubated in 25 mL of InVision His-tag In-gel Stain (Thermo Fisher Scientific, Vilnius, Lithuania) for 1 h. Finally, the gel was washed twice with 200 mL 20 mM phosphate buffer (pH 7.8) for 10 min and analysed using the Typhoon FLA 9500 laser-based scanner (GE Healthcare, Piscataway, NJ, USA) with a 532 nm green laser (excitation source) and LPG filter (575 nm long pass emission filter); [53].

### 4.8. Enzyme Activity Assay

β-1,3-glucanase and chitinase activities were assayed spectrophotometrically at 540 nm in crude protein extracts from transgenic plants and the non-transgenic control using laminarin (Sigma, St. Louis, MO, USA) and glycol chitin, respectively, as a substrate. Each reaction mixture consisted of 100 μg crude protein extracts supplemented with 2% laminarin and 1% glycol chitin in 1 mL sodium acetate buffer (pH 5.2) and incubated for 2 h. The amount of reducing sugars was measured using the 3,5-dinitrosalicylic acid (DNS) method [61]. One unit of enzyme activity (U = μmol/min) was defined as the amount of the enzyme that liberated 1 μmol reducing sugar equivalent to N-acetyl-D-glucosamine (Serva, Heidelberg, Germany) per minute. Assays were carried out in triplicate.

### 4.9. In-Vitro Antifungal Activity Assay

For antifungal assay modified protocol of Broekaert et al. [62] was used. Spores from cultures of filamentous fungi *F. poae*, *T. viride*, *A. solani,* and *R. solani* were diluted to a final spore concentration of 10^5^ spores/mL in potato dextrose broth (PDB). The cultivation mixture (1500 μL) was performed in 1.5 mL microtubes and consisted of 500 µL 50 mM acetate buffer (pH 5.2), 500 μL spore culture, and 100 µg crude protein extracts (diluted in 500 µL of acetate buffer) from transgenic plants and the non-transgenic control. Alternatively, 50 µg purified rDbGluc1 protein diluted in 500 µL acetate buffer were tested. Cultivation was performed at 25 °C for 24 h in *F. poae* and *T. viride*, 34 h in *R. solani*, and 40 h in *A. solani*. Absorbance was measured at 595 nm on a Synergy H1 microplate reader (BioTek, Winooski, VT, USA) at the start (T_0_) and end (T) of the experiment. The percentage of inhibition was calculated as A_595_ (T − T_0_) of the sample ÷ A_595_ (T − T_0_) of the control × 100. In total, four and five biological and three technical replicates were performed for the purified rDbGluc1 protein and crude protein extracts, respectively. 

### 4.10. Statistical Analyses

Data obtained from enzyme and antifungal activity assays were statistically analysed by one-way analysis of variance (ANOVA; Welch’s), followed by Games-Howell Post-Hoc Test using Jamovi software (v.1.6.15.0). Probability values less than 0.05 (*p* < 0.05) were considered significant.

## 5. Conclusions

This study was undertaken to investigate the antifungal effect of purified sundew β-1,3-glucanase and crude protein extracts isolated from transgenic tobacco plants carrying β-1,3-glucanase driven by the d*CaMV35S* promoter. The recombinant purified protein inhibited the growth of *R. solani, A. solani,* and *F. poae* but not *T. viride*. PCR and RT-PCR analyses of regenerated transgenic plants confirmed the presence of *DbGluc1* and expression of the transgene. Transgenic tobacco lines 13, 16, 19, and 22 revealed 2.7-, 3.6-, 5.9-, and 5.7-times higher transgene expression, resulting in 1.7-, 1.8-, 2.5-, and 2.3-fold increases in total β-1,3-glucanase activity, respectively, compared to the non-transgenic control. Antifungal in-vitro tests, in which crude protein extracts were supplemented to the phytopathogen cultures, showed that lines 19 and 22 were able to significantly inhibit fungal growth. These lines, unlike lines 13 and 16, increased the activity of endogenous chitinase(s) by 20%, which correlated with a significant antifungal effect. Inhibition of mycelial growth was highest for *F. poae* (28–34%), followed by *R. solani* (17%) and *A. solani* (14%). 

## Figures and Tables

**Figure 1 plants-10-01747-f001:**
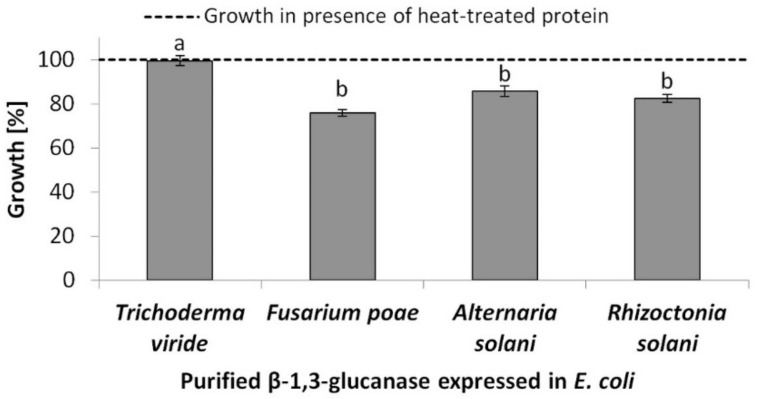
Antifungal effect of 50 µg of purified rDbGluc1 protein on the growth of filamentous fungi *Trichoderma viride*, *Fusarium poae*, *Alternaria solani*, and *Rhizoctonia solani*. A difference in absorbance at 595 nm between the start (T_0_) and final point of co-cultivation (T) compared to the control (heat-treated protein) was used. Values represent the mean of four biological and three technical replicates (Appendix A). ^a,b^ difference at *p* ˂ 0.05 compared to the control.

**Figure 2 plants-10-01747-f002:**
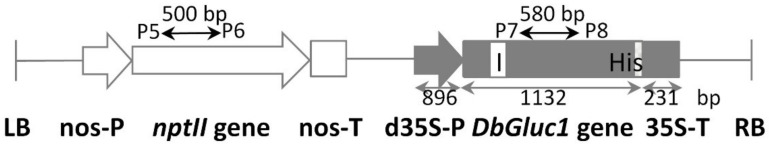
Schematic diagram of the T-DNA (transfer DNA) region of the vector construct pDbGlucBin+ (not to scale). The gene for *D. binata* β-1,3-glucanase (*DbGluc1* gene) was under the control of the double *CaMV35S* promoter (d35S-P) and terminated by the *CaMV35S* terminator (35S-T). The gene for neomycin phosphotransferase (*nptII* gene) was driven by the nopaline transferase promoter (nos-P) and *nos* terminator (nos-T) sequence. The intron and His-tag sequences are marked by (I) and (His), respectively. Black arrows indicate the position of the primers used for PCR analyses.

**Figure 3 plants-10-01747-f003:**
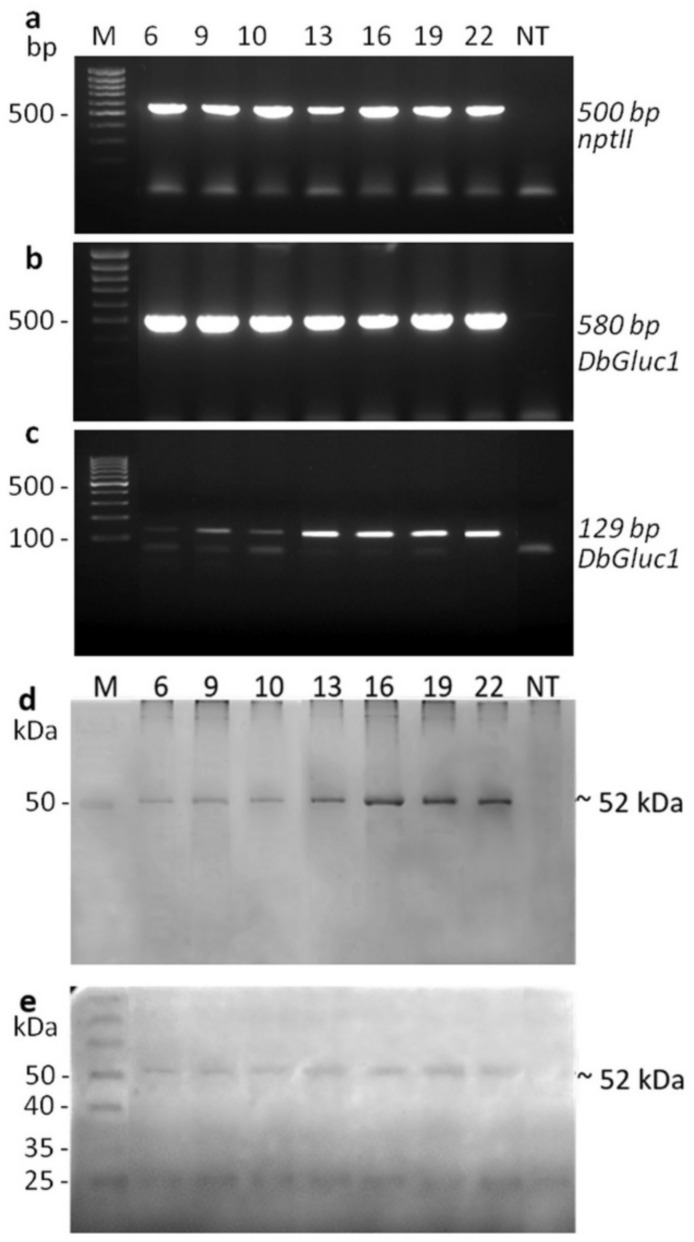
Analysis of transgene integration and expression by the PCR method (**a**,**b**); RT-PCR (**c**); the DbGluc1-HisTag detection technique (**d**) and SDS-PAGE detection of DbGluc1 activity (**e**). *DbGluc1* (**a**) and *neomycin phosphotransferase* II (**b**) genes were detected in genomic DNA of transgenic plants with specific primers as PCR products of 580 and 500 bp, respectively. (**c**) RT-PCR analysis with primers designed within the *DbGluc1* gene with the expected 129 bp length of the PCR product. (**d**) DbGluc1-His-tag protein expression examined using the InVision His-tag In-gel Stain detected as a single 52 kDa band with varying intensity in individual transformants. (**e**) Detection of β-1,3-glucanase activity on SDS-PAGE with laminarin following SDS removal.

**Figure 4 plants-10-01747-f004:**
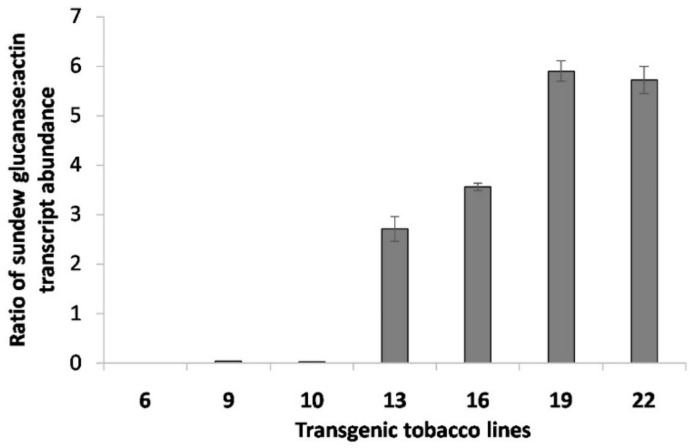
Relative expression levels of the *DbGluc1* gene in transgenic tobacco lines. Relative expression levels were calculated with the Pfaffl method as the ratio between transcript abundance of the target *DbGluc1* transgene and the endogenous *actin* gene control. Normalisation was performed based on the amplification efficiency results from a 5 point 5-fold dilution standard curve (DbGluc1: E = 1.813, R^2^ = 0.999; actin: E = 1.854, R^2^ = 0.998). Error bars are standard deviations calculated from three independent biological samples, each with technical duplicates.

**Figure 5 plants-10-01747-f005:**
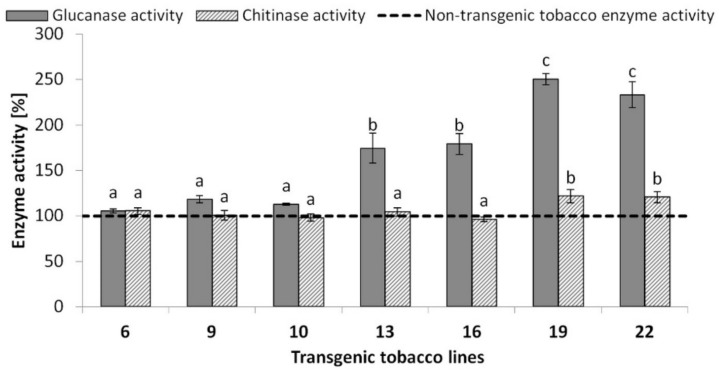
The 3,5-dinitrosalicylic acid (DNS) measurement of reducing sugars released by β-1,3-glucanase and chitinase, respectively, in crude protein extracts of transgenic tobacco lines and the non-transgenic control. ^a,b,c^ difference at *p* ˂ 0.05 from the control.

**Figure 6 plants-10-01747-f006:**
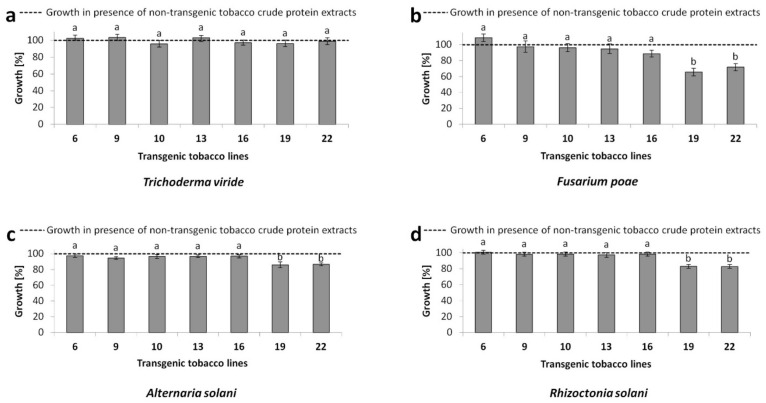
Antifungal activity effect of 100 μg of crude protein extracts isolated from transgenic tobacco lines 6, 9, 10, 13, 16, 19, and 22 and the non-transgenic control on the fungal growth of filamentous fungi (**a**) *T. viride*, (**b**) *F. poae*, (**c**) *A. solani,* and (**d**) *R. solani* grown in liquid potato dextrose broth. Values represent the mean of five biological and three technical replicates (Appendix A) ^a,b^ difference at *p* ˂ 0.05 from the control.

## Data Availability

The data presented in this study are available in the text, figures, and Appendix A.

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
