# Peer review of "Basic β-1,3-Glucanase from Drosera binata Exhibits Antifungal Potential in Transgenic Tobacco Plants"

_plants, 2021, doi:10.3390/plants10081747_

Round 1

Reviewer 1 Report

The article by Rajninec et. al. titled 'Basic β-1,3-glucanase from Drosera binata exhibits antifungal potential in transgenic plants' has highlighted the importance of heterologous expression and antifungal properties of basic β-1,3-glucanase gene from a carnivorous plant source. This study is interesting and might inspire researchers to look towards carnivorous plants for many of their unique properties to utilize.

The following comments are for authors to further improve this manuscript. 

  1. As authors have not tested other plants in their study, therefore, they should add the 'tobacco' plant in the title of this manuscript i.e. in place of 'transgenic plant' please use ' transgenic tobacco plant'.
  2. What was the size of rDbGluc1 protein purified from E. coli?
  3. Cite some suitable references for the spectrophotometric method used in this study. Please check line no. 91 and 168.

Author Response

Responses to the comments of Reviewer 1

As authors have not tested other plants in their study, therefore, they should add the 'tobacco' plant in the title of this manuscript i.e. in place of 'transgenic plant' please use ' transgenic tobacco plant'.

Response

Based on the recommendation of the Reviewer, the title of the manuscript was modified as follows:

Basic β-1,3-glucanase from Drosera binata exhibits antifungal potential in transgenic tobacco plants

What was the size of rDbGluc1 protein purified from E. coli?

Response

The theoretically calculated molecular weight of DbGluc without signal peptide was 34.7 kDa. In E. coli, the expression strain produced the protein of 38.4 kDa including the tags on both ends of recombinant protein. The His-tag on the N-terminal enabled the purification of the rDbGluc protein on Ni-NTA agarose.

The text was edited as follows:

…..Following rDbGluc1 expression and purification of 37.2 kDa recombinant protein (including the tags on both ends) on Ni-NTA agarose, 50 µg of purified protein were supplemented into liquid cultures of filamentous fungi to inhibit the biomass increase and spectrophotometrically tested…..

Cite some suitable references for the spectrophotometric method used in this study. Please check line no. 91 and 168.

Response

We have edited the missing information in the section Results original lines no 91 and 168 and section Material and Methods original line 429.

.... 50 µg of purified protein were supplemented into liquid cultures of filamentous fungi to inhibit the biomass increase and spectrophotometrically tested at 595 nm.

2.4. Quantification of β-1,3-glucanase activity

The β-1,3-glucanase activity assay was performed spectrophotometrically at 540 nm with laminarin….

 2.5. In vitro antifungal activity assay

The hyphal growth of tested filamentous fungi was evaluated spectrophotometrically at 595 nm in the presence of crude proteins (extracted in acetate buffer)

Section

4.9. In vitro antifungal activity assay

For antifungal assay modified protocol of Broekaert et al. 1990 was used. Spores……..

Reviewer 2 Report

Manuscript ID plants-1332183:

Basic β-1,3-glucanase from Drosera binata exhibits antifungal potential in transgenic plant.  

This work presents an area of great importance for obtaining transgenic plants expressing genes related to the defense against phytopathogens. A β-1,3-glucanase from Drosera binata was cloned and expressed in tobacco using bacterial and plant expression systems. The crude extract was used to determine the inhibitory effect on growth of Trichoderma viride, Rhizoctonia solani, Alternaria solani, and Fusarium poae. I find the work quite interesting technically, with promising data having been generated.  Although the data obtained in this work could contribute to the existing literature the work needs a major revision such as those cited below:

Title: the name of the transgenic plant must be included in the title.

Basic β-1,3-glucanase from Drosera binata exhibits antifungal  potential in transgenic tobacco plant.

The abstract is adequate

Introduction:  

Authors should report the state of the art of β-1,3-glucanase in tobacco plants in the introduction. There is a vast literature on this topic.

Authors should justify the choice of the fungi T. viride, R. solani, A. solani and F. poae. Was the choice based on the cell wall composition of these fungi? What are the differences in the composition of proteins and carbohydrates? Which of these fungi cause diseases in tobacco?

Line 73: Is the statement that "ß-1,3-glucanases from digestive juice are used for the digestion of cell walls of insects containing ß-1,3-glucan" correct? Which insects?

Materials and Methods

The authors have used suitable methods for the cloning and expression of β-1,3-glucanase in tobacco plants.

Line 420: The laminarin concentration should be expressed as 2% (2mg/mL) in the same way as glycol chitin.

Line 427: item 4.9. In vitro antifungal activity assay.

The spectrophotometry technique used for this test is not ideal. The fungi used have different growths on the medium used and it is difficult to compare by measuring the absorbance at 595 nm. The proper method is to evaluate growth in PDA medium containing filter-sterilized enzymes.

Results

The results obtained were well presented. Tables and figures are clear and of good quality.

In my opinion, the work lacked “in vivo”experiments to assess whether the transgenic plants were less susceptible to diseases.

Discussion

This section needs to be improved and data obtained in this work should be compared with others in the literature. The authors repeat the information already presented in the results section. In this section, the discussion about the effect of the ß-1,3-glucanases enzyme on the inhibition of fungal growth should be deepened. Why was there no inhibition of T. reesei growth? Is the cell wall composition of this fungus different from others? Does T. reesei produce proteases in the culture medium that inactivated the ß-1,3-glucanases?

Author Response

Manuscript ID plants-1332183:

Responses to the comments of Reviewer 2

Introduction:  

Authors should report the state of the art of β-1,3-glucanase in tobacco plants in the introduction. There is a vast literature on this topic.

Authors should justify the choice of the fungi T. virideR. solaniA. solani and F. poae. Was the choice based on the cell wall composition of these fungi? What are the differences in the composition of proteins and carbohydrates? Which of these fungi cause diseases in tobacco?

Response

Based on the Reviewer´s recommendation, the Introduction was heavily revised and supplemented by the state of the art of information about β-1,3-glucanases in tobacco and of tested fungi. We hope that the revised Introduction will meet the expectation of the Reviewer

The revised text of the Introduction is highlighted in red in the revised manuscript

Line 73: Is the statement that "ß-1,3-glucanases from digestive juice are used for the digestion of cell walls of insects containing ß-1,3-glucan" correct? Which insects?

Response

This passage was corrected for clarity as:

…..β-1,3-glucanases are harnessed for the digestion of trapped insect prey cell walls containing β-1,3-glucan……

Materials and Methods

Line 420: The laminarin concentration should be expressed as 2% (2mg/mL) in the same way as glycol chitin.

Response

The text was edited as follows:

…of 100 μg crude protein extracts supplemented with 2% laminarin (Sigma) and 1% glycol chitin in 1 mL sodium acetate buffer (pH 5.2…)

Line 427: item 4.9. In vitro antifungal activity assay.

The spectrophotometry technique used for this test is not ideal. The fungi used have different growths on the medium used and it is difficult to compare by measuring the absorbance at 595 nm. The proper method is to evaluate growth in PDA medium containing filter-sterilized enzymes.

Response

To quantify the antifungal properties of the rDbGluc1 enzyme, we tested several assays published for the hydrolytic enzymes—β-1,3-glucanase and chitinase—in pilot experiments. Our experience showed that the results were the most consistent when the test fungus was grown in a liquid medium containing potato dextrose broth supplemented with purified rDbGluc1 protein versus heat-inactivated purified rDbGluc1 protein (control) and the biomass increase was quantified spectrophotometrically at 595 nm.

We agree with the Reviewer that this approach enables to compare the inhibitory effect for individual fungi and not between the fungi tested because they had a different growth curve.

We believe that the test used gave consistent results, which were demonstrated by a small variation in experiments involving four biological and three technical replicates (Fig. 1).

However, we also tried a solid PDA medium for evaluation of the fungal growth. In this approach, we aseptically punched a hole with a diameter of 8 mm with a sterile cork borer and applied the purified protein mixed with fungal inoculum. The disadvantage of this approach was the limited possibility of quantification, as the protein exhibited a small inhibitory effect on the growing fungus, likely due to the accessibility of the substrate only in fungal hyphal tips. In addition, quantification performed by the mycelial diameter measurement of a growing fungus had some shortcomings, as the fungus did not always grow evenly. For these reasons, we decided to use a modified assay, where the interaction growing fungi-tested enzyme occurred in liquid medium and increase of biomass was quantified spectrophotometrically. To the best of our knowledge methods, where β-1,3-glucanases and chitinases are applied into the PDA medium before its solidification are not implemented, probably because they require a high amount of the purified enzyme.

Results

The results obtained were well presented. Tables and figures are clear and of good quality. In my opinion, the work lacked “in vivo”experiments to assess whether the transgenic plants were less susceptible to diseases.

Response

We agree with the Reviewer that in vivo tests would strengthen the evidence about the antifungal effect of DbGluc1 protein in transgenic plants. Unfortunately, current technical conditions in our laboratory do not allow performing these tests reliably. Therefore, we focused on robust and reliable in vitro tests. Our results showed that purified DbGluc1 protein, as well as extracts from transgenic tobacco plants with over-expressed sundew glucanase gene, showed apparent antifungal potential on tested phytopathogens.

Discussion

This section needs to be improved and data obtained in this work should be compared with others in the literature. The authors repeat the information already presented in the results section. In this section, the discussion about the effect of the ß-1,3-glucanases enzyme on the inhibition of fungal growth should be deepened. Why was there no inhibition of T. reesei growth? Is the cell wall composition of this fungus different from others? Does T. reesei produce proteases in the culture medium that inactivated the ß-1,3-glucanases?

Response

Discussion about the effect of the ß-1,3-glucanases on the inhibition of fungal growth was amended. It is highlighted in red in the revised version of the manuscript.

Reviewer 3 Report

Whole manuscript does satisfy the structure of writing a scientific paper. With respect to the manuscript that I reviewed entitled: Basic β-1,3-glucanase from Drosera binata exhibits antifungal potential in transgenic plantse, you will be interested in the following appreciations and no observations:   

Abstract: concise, descriptive, emphasizing the idea, aim of the present study.  

Introduction: is well written, well documented with relevant literature (very actual, from the last 10 years) and it is able to introduce and familiarize with the subject of the study, in a very gradual and logical way.

Materials and Methods: clear, logical and easily replicable. Perhaps is a little bit large, but I believe that this is the authors ‘choice to describe in very details the used methods, comparatively to others, who only refer to the standard method, without giving many technical details. However, this is not in any case a weak point, but only a simple observation.

Regarding the section results is presented in a concise, easy to follow manner, also using the 6 figures (13 graphs), which are very well organized.

The discussion section is concise and compared to the research of other authors.The statements are clearly supported by data and are linked to the paper's goal. Study implications and limitations are completely and succinctly presented.

Conclusions: concise and very well presented

References: correlated well with the text

Author Response

Responses to the comments of Reviewer 3

Comments and Suggestions for Authors

Whole manuscript does satisfy the structure of writing a scientific paper. With respect to the manuscript that I reviewed entitled: Basic β-1,3-glucanase from Drosera binata exhibits antifungal potential in transgenic plantse, you will be interested in the following appreciations and no observations:   

Abstract: concise, descriptive, emphasizing the idea, aim of the present study.  

Introduction: is well written, well documented with relevant literature (very actual, from the last 10 years) and it is able to introduce and familiarize with the subject of the study, in a very gradual and logical way.

Materials and Methods: clear, logical and easily replicable. Perhaps is a little bit large, but I believe that this is the authors ‘choice to describe in very details the used methods, comparatively to others, who only refer to the standard method, without giving many technical details. However, this is not in any case a weak point, but only a simple observation.

Regarding the section results is presented in a concise, easy to follow manner, also using the 6 figures (13 graphs), which are very well organized.

The discussion section is concise and compared to the research of other authors. The statements are clearly supported by data and are linked to the paper's goal. Study implications and limitations are completely and succinctly presented.

Conclusions: concise and very well presented

References: correlated well with the text

Response

Thank you so much for the positive opinion on our manuscript.

Round 2

Reviewer 2 Report

Most of the considerations that  I requested in the manuscript have been answered. Regarding in vivo experiments, the authors explained that they are currently unable to carry them out. Therefore, my opinion is for the article to be accepted for publication.